# Two Experts Are All You Need for Steering Thinking: Reinforcing Reasoning in MoE Models Without Additional Training

**Mengru Wang**[*,1] , **Xingyu Chen**[*,2] , **Yue Wang**[*,2] , **Zhiwei He**[*,2] , **Jiahao Xu**[2] , **Tian Liang**[2] ,
**Qiuzhi Liu**[2] , **Yunzhi Yao**[1] , **Wenxuan Wang**[2] , **Ruotian Ma**[2] , **Haitao Mi**[2] ,
**Ningyu Zhang**[†,2] , **Zhaopeng Tu**[†,2] , **Xiaolong Li**[2] , and **Dong Yu**[2]

[1]Zhejiang University
[2]Tencent

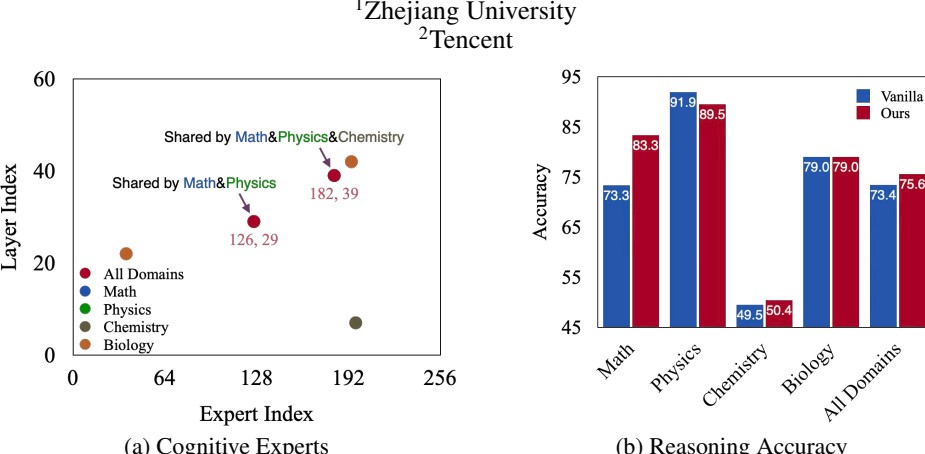

(a) Cognitive Experts    (b) Reasoning Accuracy

Figure 1: (a) Illustration of cognitive experts identified across domains. (b) Reinforcing only the top two experts (in red color) can improve reasoning accuracy without additional training.

## Abstract

Mixture-of-Experts (MoE) architectures within Large Reasoning Models (LRMs) have achieved impressive reasoning capabilities by selectively activating experts to facilitate structured cognitive processes [1, 2]. Despite notable advances, existing reasoning models often suffer from cognitive inefficiencies like overthinking [3] and underthinking [4]. To address these limitations, we introduce a novel inference-time steering methodology called **Rei**nforcing **C**ognitive **E**xperts (RICE), designed to improve reasoning depth and efficiency without additional training or complex heuristics. Leveraging normalized Pointwise Mutual Information (nPMI), we systematically identify specialized experts, termed **cognitive experts** that orchestrate meta-level reasoning operations characterized by tokens like "`<think>`". Empirical evaluations with leading MoE-based LRMs (DeepSeek-R1 and Qwen3-235B) on rigorous quantitative and scientific reasoning benchmarks (AIME and GPQA Diamond) demonstrate noticeable and consistent improvements in reasoning accuracy, cognitive efficiency, and cross-domain generalization. Crucially, our lightweight approach substantially outperforms prevalent reasoning-steering techniques, such as prompt design and decoding constraints, while preserving the model's general instruction-following skills. These results highlight reinforcing cognitive experts as a promising, practical, and interpretable direction to enhance cognitive efficiency within advanced reasoning models.

[*]Equal Contribution. Work was done when Mengru, Xingyu, Yue, and Zhiwei were interning at Tencent.
[†]Correspondence to: Zhaopeng Tu <zptu@tencent.com> and Ningyu Zhang <zhangningyu@zju.edu.cn>.

39th Conference on Neural Information Processing Systems (NeurIPS 2025).

# 1 Introduction

Models capable of extended reasoning, often referred to as Large Reasoning Models (LRMs) like OpenAI's o1 [5] and DeepSeek-R1 [1], have significantly advanced machine intelligence, largely by scaling test-time computation [6, 7]. Despite their impressive capabilities, these LRMs remain susceptible to inefficiencies [8–15]. Prior work has sought to address these issues through approaches such as preference optimization [3], decoding penalties [4], and various other techniques. In this work, we tackle these problems from a novel perspective: potential expert specialization in Mixture-of-Experts (MoE) architecture.

Due to the computational resource efficiency brought about by its sparsity, the MoE architecture has been increasingly adopted by state-of-the-art (SOTA) models, such as DeepSeek-R1 [1] and Qwen3 [2]. This sparse, specialized activation paradigm bears a conceptual resemblance to functional specialization in the human brain, where targeted interventions can modulate cognitive functions and behaviors [16–19]. Inspired by this principle, we systematically investigate whether undesirable reasoning behaviors in MoE-based LRMs correlate with the activation patterns of specific experts, and critically, if strategic manipulation of these experts can ameliorate such issues.

We introduce an approach to identify and modulate key experts integral to the reasoning process. By analyzing the co-occurrence of explicit linguistic markers of thought (e.g., "`<think>`" and "`</think>`") with individual expert activations, we pinpoint a subset of experts highly correlated with the model's cognitive deliberations. We designate these critical components as **cognitive experts**. Through extensive experimentation with SOTA MoE-reasoning models DeepSeek-R1 [1] and Qwen3-235B [2] on challenging math and scientific reasoning benchmarks, we demonstrate that selectively amplifying **as few as two cognitive experts** can enhance both reasoning depth and efficiency. Notably, our approach achieves marked accuracy improvements while reducing token usage in critical reasoning tasks, outperforming existing steering methods such as prompting and decoding constraints [4].

Moreover, we showcase impressive generalization and robustness of cognitive expert modulation, observing consistent improvements in unseen and more complex reasoning scenarios while maintaining or even enhancing general instruction-following capabilities. Our findings provide strong evidence that modulating selective experts responsible for meta-level reasoning is effective, efficient, and broadly applicable across domains, paving the way for lightweight and interpretable model steering in increasingly sophisticated MoE-based reasoning models.

Our main contributions are:

1. We propose a normalized Pointwise Mutual Information (nPMI) method for identifying cognitive experts within LRMs that are highly correlated with reasoning behavior, requiring only a single forward propagation and no additional training.

2. We introduce a lightweight inference-time steering strategy, named "reinforcing cognitive experts", that effectively enhances reasoning depth and accuracy without requiring any additional training or supervision signals.

3. Through comprehensive experiments on two prevalent MoE reasoning models and rigorous benchmarks, we empirically validate the efficacy, generalizability, and robustness of cognitive expert modulation, demonstrating significant improvements in cognitive efficiency and problem-solving accuracy.

# 2 Identifying Cognitive Experts

In this section, we leverage normalized Pointwise Mutual Information (nPMI) [20] to quantify the correlation between model thinking and each expert in a Mixture of Experts (MoE) reasoning model. We hypothesize that there are some "cognitive experts" selected by nPMI metric, which orchestrate meta-level reasoning for complex tasks.

## 2.1 Expert Specialization in MoE Models

In large reasoning models, deep thinking is manifested through key tokens, such as "`<think>`" to initiate reasoning, "`</think>`" to terminate it, and tokens like "`recheck`" to guide introspection. In

the MoE framework, these tokens are generated during forward propagation through various model components, including the MoE routing mechanism that assigns them to specialized experts, with weights determining each expert's contribution.

Formally, let us consider an MoE framework [21] with $N$ experts, denoted $\{E_1, \ldots, E_i, \ldots, E_N\}$, at each layer. For each input token $x$, a gating function selects a subset $S \subset \{E_1, \ldots, E_O\}$ of $O$ experts ($O \leq N$), where $|S| = O$, and assigns weights $w_i$ (with $\sum_{i \in S} w_i = 1$) to each selected expert $E_i \in S$. The output $h_x$ for token $x$ is computed as:

$$h_x = \sum_{i \in S} w_i \cdot E_i(x), \quad \text{where } |S| = O, \tag{1}$$

where $E_i(x)$ represents the output of expert $E_i$, and $w_i$ is the weight of the $i$-th selected expert. Prior work on MoE models shows that expert routing is often token-dependent [22], but recent study [23, 24] indicates that DeepSeek-R1's advanced reasoning enables its expert routing to focus on semantic specialization, surpassing token-dependent methods. We hypothesize that experts with consistently high co-occurrence scores with thinking tokens serve as key "cognitive experts" responsible for meta-level reasoning.

**Measuring Correlation of Specialized Experts and Thinking Tokens**   To examine whether a given expert consistently governs the model's reasoning process, we measure the co-occurrence between its activation and specific reasoning-related marker tokens, such as "`<think>`," "`</think>`", and others. Formally, let $x$ represent a token and $y$ denote expert $E_i$. We measure their association using pointwise mutual information (PMI). The PMI of $x$ and $y$ is defined as

$$\text{PMI}(x, y) = \log_2 \frac{p(x, y)}{p(x)\, p(y)} = \log_2 \frac{p(y|x)}{p(y)}, \tag{2}$$

where $p(x, y)$ is the joint probability that $x$ and $y$ both occur, while $p(x)$ and $p(y)$ are their individual (marginal) probabilities, and $p(y|x)$ is the conditional probability that $y$ occurs given $x$.

For interpretability, we normalize PMI to the range $[-1, +1]$, yielding

$$\text{nPMI}(x, y) = \frac{\text{PMI}(x, y)}{-\log_2 p(x, y)}. \tag{3}$$

Thus, $\text{nPMI}(x, y) \approx -1$ indicates that events $x$ and $y$ never co-occur, $\text{nPMI}(x, y) = 0$ implies independence, and $\text{nPMI}(x, y) \approx +1$ indicates they appear almost exclusively together (complete co-occurrence).

Let $M$ be the number of instances in a dataset, and let $T$ be the total number of tokens generated over all instances in the test set. We denote by $k_n$ the number of times the expert $E_i$ is activated specifically when the thinking token (e.g. "`<think>`") appears, and by $K_n$ the total number of times $E_i$ is activated across all tokens (including both thinking and non-thinking tokens). Since the reasoning model generally generates one thinking start and end token for each instance, then we can achieve the following functions when $x$ denotes "`<think>`" or "`</think>`":

$$p(y = E_i | x) = \frac{k_n}{M}, \qquad p(y = E_i) = \frac{K_n}{T}, \qquad p(x, y = E_i | x) = \frac{k_n}{T}. \tag{4}$$

$$\text{nPMI}(x, y = E_i) = \frac{\log_2(\frac{k_n}{M}) + \log_2(\frac{T}{K_n})}{\log_2(\frac{T}{k_n})}. \tag{5}$$

Intuitively, if an expert $E_i$ is activated almost exclusively during "`<think>`" and rarely (or never) at other tokens, $k_n \approx K_n \approx M$, $\text{nPMI}(x = \texttt{<think>}, y = E_i) \approx \frac{\log_2 1 + \log_2(\frac{T}{M})}{\log_2(\frac{T}{M})} \approx +1$, indicating that this expert is effectively tied to the thinking marker. In other words, the expert's entire usage focuses on activating the thinking token. Such specialists are prime candidates for "cognitive experts", given their consistently high co-occurrence with the thinking marker tokens.

## 2.2 Identify Cognitive Experts

We observe that some experts exhibit high nPMI scores with both "`<think>`" and "`</think>`", indicating a *bimodal association*. This suggests their broad involvement in the reasoning process

rather than specialization in its initiation. To prioritize experts specialized in initiating (rather than terminating) reasoning, we adopt the following selection strategy:

We define a set of thinking tokens $\Pi = \{\texttt{<think>}, \texttt{</think>}, \texttt{Alternatively}\}$. The normalized Pointwise Mutual Information (nPMI) score for expert $E_i$ is formulated as:

$$\text{nPMI}_{E_i} = \sum_{x \in \Pi} c_x \cdot \text{nPMI}(x, y = E_i), \tag{6}$$

where $x$ is a thinking token in set $\Pi$, $c_x$ denotes the coefficient associated with the token $x$, assigned as $c_{\texttt{<think>}} = 1$, $c_{\texttt{</think>}} = -1$, and $c_{\texttt{Alternatively}} = -1$.

Then, we select the top-$l$ experts based on their nPMI scores to form the *cognitive expert set* $P$. The weight adjustment for expert $E_i$ is governed by the following condition:

$$w_i = \begin{cases} w_i \cdot \beta & \text{if } E_i \in S \text{ and } E_i \in P, \\ w_i & \text{otherwise,} \end{cases} \tag{7}$$

where $P = \{E_j \mid \text{nPMI}_{E_j} \text{ is among the top } l \text{ scores}\}$ denotes the set of *cognitive experts*, $S$ is the subset selected by the gating function in Eq. 1, and $\beta$ is the steering multiplier. In other words, once these experts are identified, we can reinforce reasoning in the MoE model by controlling their contribution through the hyperparameter $\beta$.

# 3 Experiments

**Research Questions**    In this study, we investigate the following research questions:

RQ1: Are there "cognitive experts" specialized in thinking? If so, do these experts differ across domains?

RQ2: Can the identified cognitive experts effectively enhance cognitive effort within MoE models?

RQ3: Do "cognitive experts" differ across various domains (e.g., math, physics, chemistry, and biology)?

RQ4: Does reinforcing specific cognitive experts negatively impact the general problem-solving capabilities of MoE models?

## 3.1 Experimental Setup

**MoE-based Reasoning Models**    Currently available open-source MoE architectures tailored for large reasoning models tasks include DeepSeek-R1 [1] and Qwen3-235B [2]. DeepSeek-R1 selects 8 experts from a total of 256 at each layer, whereas Qwen3-235B selects 8 experts from a total of 128. We primarily use the DeepSeek-R1 (671B) model for our experiments, supplemented by additional evaluations on the Qwen3-235B model to examine the generalizability of cognitive experts. Note that we provide more experimental details in §B.

**Benchmarks**    We evaluate our approach on two challenging benchmarks designed specifically to test the reasoning abilities necessary for solving scientific problems across diverse domains:

- **AIME** [25]: a dataset from the American Invitational math Examination, which assesses advanced mathematical problem-solving skills. We use two recent test sets, AIME2024 and AIME2025, each comprising 30 problems.

- **GPQA Diamond** [26]: a comprehensive dataset of 198 expert-crafted multiple-choice questions in biology, chemistry, and physics, designed to test advanced scientific reasoning skills.

## 3.2 Cognitive Experts

*To address RQ1*, we first identify cognitive experts within two MoE reasoning models – DeepSeek-R1 [1] and Qwen3-235B [2] – across four scientific domains. Taking math as an illustrative example, we first use DeepSeek-R1 to generate answers on the AIME2024 dataset, simultaneously recording the expert selections at each token position during forward propagation. Next, we employ the nPMI

Table 1: Identified cognitive experts of DeepSeek-R1. Each entry (layer ID, expert ID) denotes the DeepSeek-R1 model layer ID and expert ID. "All" combines data from all domains.

| Domain | Identified Experts Ranked by nPMI Score | | | | |
| --- | --- | --- | --- | --- | --- |
| | *1st* | *2nd* | *3rd* | *4th* | *5th* |
| Math | (39, 182) | (29, 126) | (14, 114) | (27, 45) | (16, 129) |
| Physics | (29, 126) | (39, 182) | (36, 53) | (39, 46) | (24, 159) |
| Chemistry | (7, 197) | (39, 182) | (22, 37) | (29, 106) | (29, 126) |
| Biology | (42, 194) | (22, 37) | (37, 241) | (43, 61) | (39, 188) |
| All | (39, 182) | (29, 126) | (29, 106) | (4, 214) | (50, 120) |

Table 2: Identified cognitive experts of Qwen3-235B. Each entry (layer ID, expert ID) denotes the Qwen3-235B model layer ID and expert ID. "All" combines data from all domains.

| Domain | Identified Experts | | | | |
| --- | --- | --- | --- | --- | --- |
| | *Top-1* | *Top-2* | *Top-3* | *Top-4* | *Top-5* |
| Math | (70, 47) | (23, 115) | (19, 47) | (75, 46) | (22, 88) |
| Physics | (2, 28) | (74, 65) | (4, 44) | (25, 103) | (7, 36) |
| Chemistry | (32, 58) | (26, 30) | (68, 35) | (37, 57) | (25, 103) |
| Biology | (2, 28) | (26, 30) | (67, 15) | (82, 29) | (25, 103) |
| All | (25, 103) | (26, 30) | (82, 29) | (67, 15) | (37, 57) |

metric defined in Eq. 6 to identify the top five experts that exhibit the strongest statistical association with reasoning-related marker tokens (e.g., "<think>"). These experts are thus identified as the key cognitive experts specialized for mathematical reasoning. Analogously, we apply this procedure to the biology, chemistry, and physics questions in the GPQA Diamond dataset to identify cognitive experts in these respective domains. In the case of Qwen3-235B, we follow a similar procedure but generate domain-specific responses with the Qwen3-235B model itself. This ensures consistent identification signals that correspond directly to the model under examination.

We demonstrate the nPMI distribution of the top 10 experts of Deepseek-R1 across four domains (Math, Physics, Chemistry, and Biology) in Fig 2. Across the four domains, the top five experts exhibit nPMI values that are mostly above 0.5. Besides, the top 5 experts also indicate a sharply peaked distribution toward the other experts. In particular, the group of top five "thinking-specialized" experts shows significantly higher nPMI scores than the remaining experts, suggesting that domain reasoning is largely concentrated within a few highly specialized components. This pattern supports our hypothesis that a small number of experts are highly specialized for cognitive functions. Subsequently, we delve into the effectiveness of the top 5 experts in Table 3.

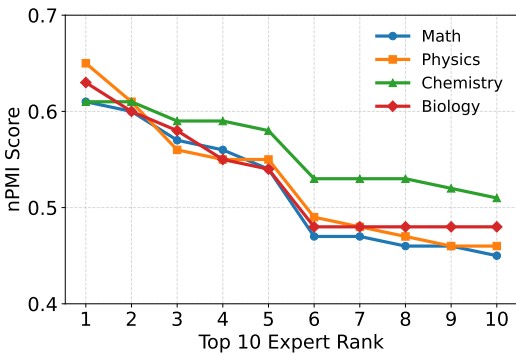

Figure 2: The nPMI distribution of top 10 expert of Deepseek-R1 across four domains.

**Cognitive Experts Across Domains**   Cognitive experts identified within DeepSeek-R1 are summarized in Table 1. An analogous summary for Qwen3-235B is provided and discussed in Table 2. From Table 1, we observe that the top two cognitive experts in the math, physics, and the aggregated "All" domains are remarkably consistent: (39, 182) and (29, 126). This strongly suggests these experts play critical and reliable roles in reasoning tasks requiring increased cognitive effort, particularly in quantitative and logic-intensive domains. The significant overlap observed between math and physics further implies a shared underlying cognitive strategy—likely focusing on symbolic manipulation and structured logical inference—which the model employs consistently across these domains. Addi-

Table 3: Effect of Deepseek-R1 on AIME24 with reinforced cognitive experts, evaluated across different multipliers and varying numbers of Math-domain cognitive experts. "Random" denotes two randomly chosen experts. The row with Multiplier 1 denotes the performance of vanilla DeepSeek-R1.

| Multiplier | Top1 | Top2 | Top3 | Top4 | Top5 | Random |
|---|---|---|---|---|---|---|
| 1 | | | 73.3 | | | |
| 2 | 70.0 | 70.0 | 76.7 | 73.3 | 73.3 | 70.0 |
| 4 | 76.7 | **83.3** | 73.3 | 66.7 | 76.7 | 73.3 |
| 8 | 76.7 | 73.3 | **83.3** | 73.3 | 73.3 | 70.0 |
| 16 | 80.0 | 80.0 | 1.7 | 76.7 | 73.3 | 73.3 |
| 32 | 70.0 | **83.3** | 73.3 | 73.3 | 73.3 | 76.7 |
| 64 | 80.0 | **83.3** | 60.0 | 53.3 | 50.0 | 66.7 |
| 128 | 70.0 | **83.3** | 43.3 | 26.7 | 13.3 | 63.3 |
| 256 | 73.3 | 60.0 | 10.0 | 6.7 | 0.0 | 73.3 |
| 512 | 63.3 | 46.7 | 6.7 | 3.3 | 0.0 | 63.3 |

tionally, the repeated appearance of certain experts in multiple domains supports our hypothesis: a subset of experts encodes generalized reasoning capabilities applicable across diverse scientific fields. Therefore, these cross-domain patterns indicate that DeepSeek-R1 may encode robust domain-general cognitive mechanisms, with some experts serving as reusable computational building blocks suitable for abstract reasoning and logical problem-solving tasks. Note that even within the same domain, there are distinctions. For instance, when comparing the top-5 cognitive experts for AIME24, MATH, and GSM8K, we find both shared and dataset-specific experts. Therefore, we hypothesize that the shared experts are responsible for general mathematical abilities, while the unique experts handle dataset-specific skills.

### 3.3 Effectiveness of Cognitive Experts

**Reinforcing Cognitive Experts**    *To answer RQ 2*, we reinforce the identified top 5 cognitive experts from the Math (AIME24) and evaluate their performance under different reinforcement configurations on the same benchmark AIME24 (Table 3). The optimal hyperparameters – the number of cognitive experts $l$ and the steering multiplier $\beta$—are selected based on this evaluation and used in all subsequent experiments. We then assess the generalization ability of these reinforced experts on the unseen, more challenging tasks from AIME25 (Table 4).

From Table 3, we observe that **reinforcing two top-ranked cognitive experts significantly enhances the model's reasoning ability**. Notably, using two experts with a steering multiplier of 4, 32, 64, or 128 achieves the highest accuracy of 83.3%. In contrast, applying an excessively large multiplier (e.g., 512) causes a dramatic drop in accuracy, often to near zero. This failure mode is characterized by the model repetitively generating meaningless tokens, suggesting that overly aggressive reinforcement disrupts the model's generation dynamics. Overall, moderate reinforcement of well-identified cognitive experts leads to consistent improvements, whereas over-reinforcement or random expert selection results in performance degradation. However, reinforcing two randomly selected experts across a wide multiplier range (2 to 512) yields minimal performance variation. Therefore, we use *two experts with a steering multiplier 64 for all subsequent experiments* [3].

We directly apply the cognitive experts identified from AIME24 to solve unseen and more challenging reasoning problems in AIME25. As shown in Table 4, these cognitive cognitive experts generalize well to the AIME25 test set. For DeepSeek-R1, the accuracy improves from 63.3% to 73.3% when guided by the identified cognitive experts. Similarly, for Qwen3-235B, accuracy increases from 66.7% to 73.3%. Additional pass@k performance using the model's officially recommended top-$p$ sampling strategy (provided in §C.4) further supports this observation. The above phenomenon demonstrates **the transferability and robustness of the expert selection across tasks with higher cognitive demands**.

---

[3]This setup is designed to test the raw generalization of the math-derived setting. However, different domains may require different intensities of cognitive steering. We discuss this in detail in §C.1.

Table 4: Performance of our approach on the AIME24 and generalization on the unseen AIME25.

| Benchmark | Method | Accuracy | Thoughts | #Tokens |
|---|---|---|---|---|
| AIME24 | DeepSeek-R1 | 73.3 | 12.0 | 9,219 |
| | +RICE {(39,182), (29,126)} | **83.3** | 10.2 | 8,317 |
| AIME25 | DeepSeek-R1 | 63.3 | 17.0 | 10,875 |
| | +RICE {(39,182), (29,126)} | **73.3** | 15.2 | 11,441 |
| AIME24 | Qwen3-235B | 86.7 | 20.1 | 10,956 |
| | +RICE {(70,47), (23,115)} | 86.7 | 16.2 | 10,722 |
| AIME25 | Qwen3-235B | 66.7 | 19.7 | 15,013 |
| | +RICE {(70,47), (23,115)} | **73.3** | 16.8 | 13,935 |

Table 5: Effect of cognitive experts of Deepseek-R1 across different domains.

| Domain | Math | Physics | Chemistry | Biology | Average |
|---|---|---|---|---|---|
| R1 | 73.3 | 91.9 | 49.5 | 79.0 | 73.4 |
| Math | **83.3** | 89.5 | 50.4 | 79.0 | **75.6** |
| Physics | **83.3** | 89.5 | 50.4 | 79.0 | **75.6** |
| Chemistry | 80.0 | **95.4** | **52.7** | 68.4 | 74.1 |
| Biology | 73.3 | 93.0 | 47.3 | 73.9 | 71.9 |
| All | **83.3** | 89.5 | 50.4 | 79.0 | **75.6** |

Crucially, the observed accuracy improvements do not necessarily entail increased computational cost in terms of token usage, supporting our hypothesis that our method encourages deeper thinking rather than just longer outputs. Our cognitive expert strategy, despite improving average accuracy of Deepseek-R1 on AIME24, uses more efficient reasoning thought [4] (10.2 vs 12.0) and tokens (8,317 vs 9,219) on average compared to the baseline. This efficiency phenomenon is also observed in Qwen3-2-35B, where the substantial accuracy gain (+6.6%) is accompanied by a notable reduction in thought (16.8 vs 19.7) and token count (13,935 vs 15,013). This suggests that **reinforcing cognitive experts helps the model to reason more effectively**, focusing computational effort more productively within the reasoning process without generating excessive verbosity. The reasoning effectiveness can be clearly observed in Table 8, where our RICE demonstrates deeper and more consistent reasoning, leading directly to the correct answer. In contrast, vanilla DeepSeek-R1 exhibits more frequent shifts in reasoning and fails to commit to its initially correct deductions.

### 3.4 Performance of Cognitive Experts across Domains

*To address RQ3*, we evaluate the transferability of domain-specific cognitive experts by applying expert sets identified from one domain to others. As the top-2 experts selected from *Math*, *Physics*, and the *All* domains are identical, their results are the same across domains. As shown in Table 5, we have several observations:

**Cognitive experts generalize well across domains.** Our evaluation, summarized in Table 5, clearly illustrates the efficacy of the identified cognitive experts in enhancing the DeepSeek-R1 model's reasoning capability across multiple domains. Leveraging cognitive experts identified from aggregated data ("All" domains) shows marked overall improvement, raising the average accuracy from 73.4% to 75.6%. Notably, substantial improvement is observed in the math tasks (from 73.3% to 83.3%). Moderate accuracy gains are also seen in Chemistry (from 49.5% to 50.4%) and minor degradation observed in Physics (from 91.9% to 89.5%), indicating broad applicability and effectiveness of these general reasoning modulators across diverse problem sets. Biology tasks show stable performance, unaffected by general expert modulation.

---

[4]We use the underthinking score from prior work [4] to quantify reasoning efficiency, with lower Thought values indicating greater efficiency.

**Domain-specific expert sets provide targeted gains.** Further analysis demonstrates the nuanced implications of domain-specific cognitive experts. Chemistry-identified experts outperform general experts significantly within their native Chemistry domain (49.5% to 52.7%) and notably enhance Physics performance (91.9% to 95.4%), highlighting potential cross-domain synergies between physics and chemistry reasoning processes. However, this specialization lowers the accuracy in math (from 83.3% with general experts to 80.0%) and more substantially limits the Biology domain performance (from 79.0% to 68.4%). Similarly, Biology-derived experts enhance task-specific results (from 91.9% to 93.0% in Physics) but degrade average performance across other domains, indicating further that specialized expert selections may negatively impact general cognitive reasoning by reinforcing overly specialized activations.

**No evidence of harmful side-effects on other domains.** Our experimental findings clearly confirm that cognitive experts, either chosen from aggregated cross-domain data or specific domains, constitute effective cognitive modulators that enhance model reasoning accuracy and efficiency. General-purpose expert adjustments deliver robust cross-domain improvements, demonstrating their fundamental importance to reasoning processes regardless of subject matter. Meanwhile, domain-specialized expert modulation illustrates substantial potential for targeted cognitive improvements, particularly within closely related scientific domains. Together, these insights validate our proposed approach as versatile, effective, and immediately deployable for enhancing efficiency, accuracy, and overall reasoning proficiency of existing MoE-based large reasoning models.

### 3.5 Impact of Reinforced Cognitive Experts on General Capabilities

*To address RQ4*, we investigate whether reinforcing cognitive experts negatively impacts the model's general capabilities, such as instruction-following. To this end, we evaluate reinforced models on the ArenaHard benchmark [27] to assess potential adverse impacts on general capabilities. The Arena-Hard benchmark, designed to evaluate instruction-following capabilities, comprises 500 challenging user queries spanning diverse scenarios. We randomly select 50 user queries as the test data and employ GPT-4-Turbo to judge pairwise comparisons of outputs against the GPT-4-0613 baseline.

**Reinforcing cognitive experts maintains or slightly improves general instruction-following capabilities.** Our experimental evaluation on the ArenaHard benchmark demonstrates that reinforcing the identified cognitive experts does not adversely impact the model's capability to handle general, challenging instruction-following tasks. As shown in Table 6, models modulated by cognitive experts derived from each domain consistently maintain or marginally improve upon the baseline DeepSeek-R1 accuracy of 91%. Specifically, the domain-specific cognitive experts from Chemistry and Biology show notable accuracy enhancements (from 91.0% to 94.0% in Chemistry; from 91.0% to 93.0% in Biology), underscoring the po-

Table 6: Effect of reinforced cognitive experts of Deepseek-R1 on ArenaHard.

| Method | Accuracy | Token |
|---|---|---|
| Vanilla | 91.0 | 2,919 |
| *Reinforce Experts from different domains* | | |
| Math | 92.0 | 2,933 |
| Physics | 92.0 | 2,933 |
| Chemistry | 94.0 | 3,332 |
| Biology | 93.0 | 3,072 |
| All | 92.0 | 2,933 |

tential for positive transfer of reasoning-rich expert reinforcement to general-purpose capabilities. Moreover, the general experts ("All" domain) also marginally improve performance (to 92.0%), confirming that cognitive expert-control has a neutral-to-beneficial impact on general instruction-following capabilities.

**Modulation of cognitive experts results in moderately increased verbosity.** An analysis of token counts further reveals that cognitive expert modulation moderately increases model verbosity in response generation, suggesting enhanced cognitive thoroughness. For example, Chemistry and Biology models increase average token counts notably (from 2,919 to 3,332 tokens and from 2,919 to 3,072 tokens, respectively), highlighting that the activation of certain domain-specific cognitive experts may favor more detailed deliberations. Nevertheless, the overall increase in verbosity is moderate, indicating a desirable balance between detail-oriented reasoning and response conciseness.

**Overall, reinforcing cognitive experts does not hinder but rather supports general capabilities.** These findings collectively confirm our approach as effective and safe for targeted, lightweight interventions. Reinforcing cognitive experts significantly enhances model performance within their

original domains and has either neutral or positive effects on general-purpose instruction-following benchmarks. The moderate increase in verbosity indicates richer, more thoughtful reasoning, aligning with the intended goal of encouraging deeper cognitive processing without sacrificing practicality. This highlights the practicality and versatility of our approach in improving existing MoE model reasoning efficacy and general cognitive capabilities through strategic expert modulation.

### 3.6 Comparison with Other Methods

We compare our cognitive experts against two prevalent inference-time methods for reasoning tasks: prompt engineering and decoding constraints. Specifically, we analyze two prompt configurations: placing the prompt before the `<think>` token ($\text{Prompt}_{\text{before}}$) and after `<think>` token ($\text{Prompt}_{\text{after}}$), with details outlined in Appendix B.1. For decoding constraints, we adopt a strategy similar to TIP from Wang et al. [4], which curtail the generation of alternative solutions to foster coherent and focused reasoning. In our work, we penalize the think mark tokens (`</think>`) rather than the thought switching tokens (e.g., "alternatively"), and we name the method as $\text{TIP}_t$.

Table 7 compares our cognitive expert modulation method against prompting (both before and after the `<think>` token) and decoding constraints (TIP) on the challenging AIME benchmarks. Our approach achieves the highest average improvement, boosting DeepSeek-R1's performance from a baseline of 68.3% to 78.7%, surpassing the best-performing baseline $\text{TIP}_t$ by 2.0% in average accuracy. Notably, our method achieves particularly pronounced gains on AIME24 (83.3%) compared to $\text{Prompt}_{\text{before}}$

Table 7: Comparison with other steering methods on AIME24 and AIME25.

| Method | AIME24 | AIME25 | Avg. |
|---|---|---|---|
| DeepSeek-R1 | 73.3 | 63.3 | 68.3 |
| $\text{Prompt}_{\text{before}}$ | 76.7 | 73.3 | 75.0 |
| $\text{Prompt}_{\text{after}}$ | 76.7 | 66.7 | 71.7 |
| $\text{TIP}_t$ | 80.0 | 73.3 | 76.7 |
| RICE | 83.3 | 73.3 | **78.7** |

(76.7%) and $\text{TIP}_t$ (80.0%), indicating a superior ability to enhance cognitive effort and deep reasoning through targeted modulation of expert routing weights. These results validate our hypothesis that strategically steering cognitive experts significantly reinforces appropriate cognitive behavior and enhances reasoning accuracy more effectively than generic prompting and decoding constraints.

Table 8: An example of our RICE corrects wrong reasoning (in red) generated by Deepseek-R1 into correct reasoning (in green).

| | |
|---|---|
| Problem | Let ABCDEF be a convex equilateral hexagon in... and EF has side lengths 200, 240, and 300. Find the side length of the hexagon. |
| DeepSeek-R1 | Okay, so I need to find the side length...the triangle formed by extending AB, CD, and EF has sides 200, 240, and 300. Let me sketch this mentally...**Alternatively**, maybe we can model the hexagon as being formed by three pairs of parallel sides...**Alternatively**, think of the hexagon as a combination of three parallelograms...**Alternatively**, using the concept of similar triangles...**Alternatively**, maybe each pair of opposite sides of the hexagon is parallel to a different side of the triangle...**I will posit that the answer is 80**...Therefore, **the answer is 75/2**...Let me **verify** this result...Thus, s = 1200 / 23. Wait, this **contradicts** our previous result...**I think the answer is 80**...based on the thorough step-by-step analysis, $\boxed{1200/23}$ is the conclusion. |
| +RICE | Okay, so I need to find the side length...Let me draw a rough sketch in my mind...**So, the triangle is such that each side is aligned with one of the extended sides of the hexagon**...But maybe using **coordinate geometry could help here**....Not sure, maybe this is **getting too complicated**...**Alternatively**, maybe using ratios...**Alternatively**, after struggling through all these approaches, perhaps the answer is related to the harmonic mean of the triangle's sides...Therefore, I think **the answer is 80. But need to verify**...Therefore, the side length of the hexagon is $\boxed{80}$. |

## 4 Related Work

**Large Reasoning Models** Large Reasoning Models (LRMs) significantly enhances the reasoning capabilities of large language models (LLMs) [5, 28]. Prominent implementations include OpenAI's

o1 [5], QwQ [29], Qwen3 [2], DeepSeek-R1 [1], Claude 3.7 [30] and Kimi-1.5 [31] achieve human-like reasoning by leveraging scaled test-time computation. In particular, the open-source DeepSeek-R1 utilizes a Mixture-of-Experts (MoE) architecture [32] with sparsely activated parameters, selectively activating only 8 out of 256 experts per layer [21]. This MoE architecture has been widely adopted in recent LLMs [33–36], achieving an optimal balance between computational efficiency and competitive performance in complex reasoning tasks.

**MoE Models**   Previous research on Mixture of Experts (MoE) models indicates that expert routing is primarily token-dependent [22]. However, Olson et al. [23] demonstrate that DeepSeek-R1's advanced reasoning capabilities enable its routing mechanism to achieve greater semantic specialization and structured cognitive processing, representing a substantial advancement over prior MoE models. Subsequently, Hazra et al. [37] train sparse autoencoders (SAEs) on DeepSeek-R1, identifying interpretable features such as backtracking, division, and rapid response patterns within the SAEs space. However, training SAEs is computationally intensive, posing significant resource demands. We employ the normalized Pointwise Mutual Information (nPMI) metric to evaluate expert specialization, requiring only a single forward propagation.

**Efficient Thinking**   Despite significant advancements, o1-like models continue to encounter substantial cognitive challenges, such as the overthinking [3, 38–40] and underthinking phenomenon [4, 11, 41]. Subsequent efforts address these issues through rule-based stop, decoding constraints [42, 4, 43–49], steering vectors [50–53], and parameters tuning [54, 3, 55–57]. There are also some works specifically designed to improve reasoning capabilities in MoE architectures by re-mixing experts through gradient-based optimization [58] or by expert pruning via sparse dictionary learning [59]. However, the resource-intensive nature of expert re-mixing algorithms makes them impractical to scale to large models such as 600B-parameter systems, whereas our method is lightweight and directly applicable to such large-scale settings. Generally, in contrast to the above strategies that primarily rely on crafted rules, extensive labeled data, or computationally expensive parameter training, our *reinforcing cognitive experts* approach achieves more efficient and deeper reasoning with only a single forward pass, without requiring any supervision signals or additional training.

## 5   Conclusion and Future Work

In this work, we investigate cognitive experts in MoE-based language models and propose an efficient nPMI-based method to identify those most relevant to reasoning. We show that steering these experts enables control over the model's reasoning with minimal computational overhead. Notably, these experts exhibit strong transferability across scientific domains, suggesting a generalizable cognitive function. Future directions include deeper investigations into the structural properties and broader applicability of cognitive experts, as well as integration with other cognitive control strategies to further enhance reasoning robustness. By uncovering this hidden layer of functional specialization within MoE models, we may open new avenues for fine-grained control over neural reasoning processes, more closely mirroring the modularity observed in biological cognitive systems.

## 6   Limitations and Broader Impacts

The internal coordination mechanisms of long-range reasoning models are inherently complex, and our nPMI-based approach may not fully capture all relevant interactions. Future work should explore more sophisticated metrics for expert identification. Besides, our validation was constrained by the current availability of open-source MoE architectures designed for long-range reasoning, limited to DeepSeek-R1 [1] and Qwen3-235B [2]. Additional testing across more diverse architectures is warranted. The ability to precisely control reasoning processes in large language models has significant implications for both AI safety and efficiency. Our method's minimal computational overhead makes it particularly promising for real-world applications where resource constraints are critical. The observed cross-domain transferability of cognitive experts suggests exciting possibilities for developing more general and adaptable AI systems.

## Acknowledgments

This work was supported by the National Natural Science Foundation of China (No. 62576307), Ningbo Natural Science Foundation (2024J020), Yongjiang Talent Introduction Programme (2021A156-G), Tencent AI Lab Rhino-Bird Focused Research Program (RBFR2024003), and Information Technology Center and State Key Lab of CAD&CG, Zhejiang University.

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

# A Renormalization

We investigate the DeepSeek Mixture-of-Experts (MoE) architecture, where each token selects 8 of 256 experts, with weights normalized to sum to 1. We examine steering specific expert weights under two conditions: with and without renormalization. The effects of the steering coefficient (reinforce factor) are presented in Table 9, with generalization performance analyzed in Table 10.

Table 9 evaluates the reinforce factor's effect on two cognitive experts. Without renormalization, accuracy peaks at 83.3% (factors 4, 32, 64, 128) but drops to 3.3% at 2048, with erratic token counts (e.g., 16,836). With renormalization, accuracy remains stable (73.3–83.3%) across most factors, with token counts varying moderately (8,383–9,508), though it declines to 66.7% at factor 256. Renormalization thus enhances robustness at higher steering coefficients.

We evaluate the generalization performance of cognitive experts, identified using normalized Point-wise Mutual Information (nPMI) within Mixture-of-Experts (MoE)-based large reasoning models, comparing three strategies: Vanilla R1, Renormalized, and Without Renormalized (wo/Renormalized). Table 10 reports performance across AIME25, Physics, Chemistry, Biology, and their average for experts selected from AIME24.

The wo/Renormalized strategy demonstrates superior generalization, achieving an average score of 73.1, compared to 70.9 for Vanilla R1 and 68.8 for Renormalized. This 4.3-point improvement over Renormalized is driven by notable gains in AIME25 (73.3 vs. 63.3) and Biology (79.0 vs. 68.4). In Physics, Vanilla R1 (91.9) outperforms wo/Renormalized (89.5, -2.4), while in Chemistry, Renormalized (52.7) surpasses wo/Renormalized (50.4), indicating domain-specific trade-offs.

Non-renormalization avoids the computational overhead of normalization (e.g., softmax scaling of expert weights), aligning with its reported efficiency. These results confirm that non-renormalization enhances generalization in cognitive experts, offering a computationally lightweight approach to optimizing reasoning in MoE architectures.

Table 9: Reinforce factor effects of two cognitive experts with/without renormalization

| Reinforce Factor | wo/Renormalization | | Renormalization | |
|---|---|---|---|---|
| | *Acc* | *Token* | *Acc* | *Token* |
| 1 (R1) | 73.3 | 9,291 | 73.3 | 9,291 |
| 2 | 70.0 | 9,103 | 80.0 | 8,463 |
| 4 | 83.3 | 8,145 | 80.0 | 8,383 |
| 8 | 73.3 | 9,502 | 70.0 | 8,818 |
| 16 | 80.0 | 8,493 | 73.3 | 9,133 |
| 32 | 83.3 | 8,337 | 83.3 | 8,956 |
| 64 | 83.3 | 8,317 | 80.0 | 9,508 |
| 128 | 83.3 | 9,490 | 73.3 | 9,091 |
| 256 | 60.0 | 7,986 | 66.7 | 8,719 |
| 512 | 46.7 | 6,270 | 80.0 | 8,786 |
| 1024 | 23.3 | 4,378 | 73.3 | 8,564 |

Table 10: Generalization capacity of two cognitive experts selected from AIME24, with or without renormalization.

| Strategy | AIME25 | Physics | Chemistry | Biology | Average |
|---|---|---|---|---|---|
| Vanilla R1 | 63.3 | 91.9 | 49.5 | 79.0 | 70.9 |
| Renormalized | 63.3 | 90.7 | 52.7 | 68.4 | 68.8 |
| wo/Renormalized | 73.3 | 89.5 | 50.4 | 79.0 | 73.1 |

Table 11: Pass@k performance of our cognitive experts on Deepseek-R1 and Qwen3-235B-A22B. For each problem, we generated 16 responses with a temperature of 0.6 and a top p value of 0.95.

| Model | Strategy | Accuracy | | Tokens |
|---|---|---|---|---|
| | | Pass@1 | Pass@8 | |
| | **AIME24** | | | |
| Deepseek-R1 | Vanilla | 74.8 | 88.3 | 9,219 |
| | Our | 76.0 | 89.2 | 8,317 |
| | **AIME25** | | | |
| | Vanilla | 68.5 | 84.7 | 10,875 |
| | Our | 67.7 | 86.3 | 11,441 |
| | **AIME24** | | | |
| Qwen3-235B-A22B | Vanilla | 84.0 | 93.0 | 10,946 |
| | Our | 85.0 | 91.6 | 10,706 |
| | **AIME25** | | | |
| | Vanilla | 82.7 | 88.3 | 12,546 |
| | Our | 82.1 | 89.7 | 12,373 |

# B Experiment Setup

## B.1 Baselines

We evaluate our cognitive experts in comparison with two widely used inference-time techniques for reasoning tasks: prompt engineering. In particular, we consider two types of prompt placements in our analysis — one positioned before the `<think>` token (Promptbefore) and the other placed after it (Promptafter), defined as follows:

---

**Prompt before `<think>`**

<|begin_of_sentence|><|User|> <context>
You are an expert math-solving assistant who prioritizes clear, concise solutions. You solve problems in a single thought process, ensuring accuracy and efficiency. You seek clarification when needed and respect user preferences even if they are unconventional.
</context>

<solving_rules>
- Try to complete every idea you think of and don't give up halfway
- Don't skip steps
- Display solution process clearly
- Ask for clarification on ambiguity
</solving_rules>

<format_rules>
- Use equations and explanations for clarity
- Keep responses brief but complete
- Provide step-by-step reasoning if needed
</format_rules>

PROBLEM: {problem}

OUTPUT: Please think carefully and follow above rules to get the correct answer for PROBLEM. Focus on clear, concise solutions while maintaining a helpful, accurate style.<|Assistant|> <think> \n

---

Table 12: Performance of domain-specific steering multipliers across scientific domains. The math domain is evaluated using the AIME25 benchmark.

| Model | Math | Physics | Chemistry | Biology | Average |
|---|---|---|---|---|---|
| DeepSeek-R1 | 63.3 | 91.9 | 49.5 | 79.0 | 70.9 |
| DeepSeek-R1 + RICE | **73.3** | **93.0** | **54.8** | 79.0 | **75.0** |
| Qwen-235B | 66.7 | 90.7 | 49.5 | 78.9 | 71.5 |
| Qwen-235B + RICE | **73.3** | **95.3** | 49.5 | **84.2** | **75.6** |

---

**Prompt after `<think>`**

<|begin_of_sentence|><|User|> <context>
You are an expert math-solving assistant who prioritizes clear, concise solutions. You solve problems in a single thought process, ensuring accuracy and efficiency. You seek clarification when needed and respect user preferences even if they are unconventional.
</context>

PROBLEM: {problem}

<think> \n

Please think carefully and follow these rules to find the correct answer for PROBLEM.

<solving_rules>
- Try to complete every idea you think of and don't give up halfway
- Don't skip steps
- Display solution process clearly
- Ask for clarification on ambiguity
</solving_rules>

<format_rules>
- Use equations and explanations for clarity
- Keep responses brief but complete
- Provide step-by-step reasoning if needed
</format_rules>

Focus on clear, concise solutions while maintaining a helpful and accurate style.

OUTPUT:

---

## B.2 Experiments Compute Resources

We conduct our DeepSeek-R1 experiments on 16 H20 GPUs using `vllm==0.7.0`. It is worth noting that for experiments on the Qwen3-235B-A22B model, we use `vllm==0.8.5.post` because the recently released Qwen3-235B-A22B models are only compatible with `vllm` versions $\geq$ 0.8.5.

# C  Experiment Details and Results

## C.1  Steering Multiplier

We use a simple, domain-specific steering multiplier (selected from a small set of 16, 32, 64), RICE delivers consistent and significant improvements across all domains.

Table 13: Performance comparison of Deepseek-R1 with and without RICE across different datasets.

| Dataset | Model | Accuracy | #Token |
|---------|-------|----------|--------|
| GSM8K | Deepseek-R1 | 95.9 | 1028 |
| | Deepseek-R1 + RICE (Ours) | **96.0** | **1001** |
| MATH-500 | Deepseek-R1 | 95.0 | 3282 |
| | Deepseek-R1 + RICE (Ours) | **96.4** | **3204** |
| HLE | Deepseek-R1 | 4.0 | 9433 |
| | Deepseek-R1 + RICE (Ours) | **6.0** | 9445 |

## C.2 Effects on Other Datasets

We delve into the effect of these two experts identified by AIME24 on three additional diverse benchmarks: GSM8K (grade-school math), MATH (competition math), and HLE (Humanity's Last Exam, covering social science, CS, etc.). Note that we randomly sampled 100 text instances as the test set due to resource constraints. As shown in Table 13, RICE consistently improves Deepseek-R1 across all datasets, providing modest gains on high-performing tasks (GSM8K, MATH-500) and larger relative improvements on more challenging tasks (HLE).

Moreover, we compare the differences with and without RICE. Specifically, we focus on the token distribution during model decoding. We observe that tokens related to "think," "best," "good," and similar concepts are ranked higher (positioned closer to the top 1) during decoding after expert reweighting.

## C.3 Cognitive Experts of Qwen3-235B

As a case study in math, we employ Qwen3-235B to generate responses on the AIME2024 dataset, while recording the expert assignments at each token during the forward pass. Subsequently, we apply the nPMI measure defined in Eq. 6 to identify the top five experts that exhibit the highest statistical dependence on reasoning-related indicators, such as the "<think>" token. These selected experts are thus regarded as the core cognitive components specialized in mathematical reasoning. Due to computational constraints, our quantitative analysis in Table 2 focuses specifically on math-domain experts. This focused approach allows for deeper investigation of expert specialization patterns while maintaining feasible resource requirements.

## C.4 Pass@k Performance of Cognitive Experts

Table 11 presents the Pass@$k$ performance of our cognitive expert modulation approach compared to vanilla baselines across two model architectures. On DeepSeek-R1, our method demonstrates consistent improvements in Pass@8 accuracy (+0.9% on AIME24 and +1.6% on AIME25) despite showing marginal variations in Pass@1 performance. Notably, we observe a 9.8% reduction in token consumption for AIME24 while maintaining superior accuracy, suggesting improved reasoning efficiency. For Qwen3-235B-A22B, our approach achieves higher Pass@1 accuracy (+1.0% on AIME24) while showing competitive Pass@8 performance (±1.4% across datasets), with consistent reductions in computational cost (2.2% fewer tokens on AIME24 and 1.4% fewer on AIME25). The observed trade-offs between Pass@1 and Pass@8 metrics suggest that our method enhances *reliable* reasoning (as reflected in Pass@8) more than *peak* performance (Pass@1), particularly in the more challenging AIME25 benchmark. These results substantiate our hypothesis that targeted expert modulation can improve reasoning efficiency without compromising solution quality.

