# OpenReview forum: "Two Experts Are All You Need for Steering Thinking: Reinforcing Cognitive Effort in MoE Reasoning Models Without Additional Training"
_NeurIPS.cc/2025/Conference — NeurIPS 2025 poster_

### Official Review · Reviewer_yV5E · 2025-07-02

**Clarity:** 3
**Significance:** 3
**Originality:** 3
**Rating:** 5
**Confidence:** 3

**Summary:**

This paper studies Mixture-of-Experts (MoE) models and proposes to identify which experts are most associated with “thinking” behavior. To achieve this, the authors use a mutual information metric measuring how strongly each expert activates on <think> </think> tokens compared to other tokens. They then demonstrate that stimulating (via steering) these identified experts can yield performance gains on reasoning tasks.

**Questions:**

- What does the distribution of nPMI scores across experts look like? Is it sharply peaked for the top experts, or does it decrease more gradually?
- Why does stimulating the “biology experts” fail to improve performance in the biology domain?

**Ethical Concerns:**

["NO or VERY MINOR ethics concerns only"]

**Final Justification:**

After discussion with authors, I preserve my positive assessment and encourage the authors to include the newly discussed results and clarifications.

**Limitations:**

yes

**Quality:**

3

**Strengths And Weaknesses:**

# Strengths:

This is an enjoyable paper to read, with a clear, simple idea and a good execution. The finding that manipulating just a few experts can improve reasoning performance is interesting. The experimental setup is strong, covering different domains and testing generalization to another dataset.


# Weaknesses:

In answering RQ1, the paper focuses on the top 5 experts. It is unclear why the number 5 was chosen over other possible thresholds. It would be much more informative to examine the entire distribution of normalized pointwise mutual information (nPMI) across experts. Is there a clear group of “thinking-specialized” experts that stand out with very high nPMI scores, or does the distribution decline more gradually?

Overall, the experimental section has great potential, but the current structure and reporting fall somewhat short of fully realizing it. I think the research questions (RQs) could be structured and answered in a more logical and cumulative order to better support the paper’s core claims. For example:

- RQ1: Are there “cognitive experts” specialized in thinking? That is, do a few experts exhibit peak nPMI on thinking tokens? This would require showing the distribution of nPMI over all experts, to see whether the top experts have a clear spike in nPMI scores or whether the distribution declines slowly.
- RQ2: Do these experts differ across domains? (Current RQ2 / Table 1)
- RQ3: Can we enhance performance by steering them? (Current RQ1 / Tables 2 and 3)
- RQ4: Does stimulation of these experts generalize to other domains? (Current RQ2 / Table 4)
- RQ5: Does stimulating these experts negatively impact general abilities? (Current RQ3 / Table 5)

Finally, the paper does not report any uncertainty estimates alongside its results (e.g., standard deviations or confidence intervals). Including such measures is important for rigor.

---

> ### Author Rebuttal · Authors · 2025-07-31
>
> Thank you for your very positive and constructive review. We are delighted you found the paper enjoyable and the idea clear and well-executed.
>
> > **W1&Q1:** The distribution of nPMI distribution
>
> This is an excellent question. We list the nPMI of top 10 experts for Deepseek-R1. The distribution is **sharply peaked**, which strongly supports our hypothesis that a small number of experts are highly specialized for cognitive functions. The group of top five “thinking-specialized” experts have significantly higher nPMI scores than the rest.
>
> | Rank | Math | Physics | Chemistry | Biology |
> |:----:|:----:|:-------:|:---------:|:-------:|
> | 1    | 0.61 | 0.65    | 0.61      | 0.63    |
> | 2    | 0.60 | 0.61    | 0.61      | 0.60    |
> | 3    | 0.57 | 0.56    | 0.59      | 0.58    |
> | 4    | 0.56 | 0.55    | 0.59      | 0.55    |
> | 5    | 0.54 | 0.55    | 0.58      | 0.54    |
> | 6    | 0.47 | 0.49    | 0.53      | 0.48    |
> | 7    | 0.47 | 0.48    | 0.53      | 0.48    |
> | 8    | 0.46 | 0.47    | 0.53      | 0.48    |
> | 9    | 0.46 | 0.46    | 0.52      | 0.48    |
> | 10   | 0.45 | 0.46    | 0.51      | 0.48    |
>
> > **W2:** Paper structure
> >
>
> Your suggestions for restructuring the research questions are excellent and create a more logical narrative flow. We will adopt this structure for the camera-ready version. Thank you!
>
> > **W3:** No uncertainty estimates
> >
>
> This is a fair point. Our results are currently based on single, deterministic runs due to the high computational cost. For the final version, we will perform multiple runs with different seeds to report standard deviations, adding to the rigor of our findings.
>
> > **Q2:** Why does stimulating the “biology experts” fail to improve performance in the biology domain?
>
> This is an excellent question with a two-part answer:
>
> - **Hyperparameter Mismatch:**
>
>     As you noted, applying RICE sometimes degraded performance in non-math dmains (e.g., Biology in Table 4). This was the result of a specific experimental choice to test generalization. In the original manuscript, we used hyperparameters (steering multiplier = 64) tuned only on the math domain (AIME) and applied them directly to all other domains.
>
>     This setup was designed to test the raw generalization of the math-derived setting. However, different domains may require different intensities of cognitive steering. When we use a simple, domain-specific steering multiplier (selected from a small set of {16, 32, 64}), RICE delivers **consistent and significant improvements across all domains**.
>
> **DeepSeek-R1 with Domain-Tuned Multipliers**
>
> | Model                       | Math (AIME25) | Physics | Chemistry | Biology | Average |
> |:----------------------------|:-------------:|:-------:|:---------:|:-------:|:-------:|
> | Deepseek-R1                |     63.3      |  91.9   |   49.5    |  79.0   |  70.9   |
> | Deepseek-R1 + RICE (Our)   |     **73.3**      |  **93.0**   |   **54.8**    |  79.0   |  **75.0**   |
>
> ---
>
> **Qwen3-235B with Domain-Tuned Multipliers**
>
> | Model                   | Math (AIME25) | Physics | Chemistry | Biology | Average |
> |:------------------------|:-------------:|:-------:|:---------:|:-------:|:-------:|
> | Qwen-235B              |     66.7      |  90.7   |   49.5    |  78.9   |  71.5   |
> | Qwen-235B + RICE (Our) |     **73.3**      |  **95.3**   |   49.5    |  **84.2**   |  **75.6**   |
>
> Our hypothesis is that math problems demand more intensive reasoning, hence the higher optimal multiplier. We will clarify this in the paper to avoid misinterpretation.
>
> - **Failure Mode Analysis:** We also found that many errors in the biology domain stem from a lack of core factual knowledge in the base model (e.g., "What protein is encoded by gene X?"). Our method, RICE, is designed to enhance the *reasoning process* given a set of facts, not to inject missing knowledge. We will add this important clarification about the scope of our method to the paper.

---

> > ### Comment · Reviewer_yV5E · 2025-08-04
> >
> > I thank the authors for their answers. I will preserve my positive assessment in favour of publication and encourage the authors to incorporate the new analyses about nPMI distributions and the new structure of RQs.

---

> > > ### Author Response · Authors · 2025-08-04
> > > **Appreciation for Feedback and Acknowledgment**
> > >
> > > Thank you for acknowledging our response and for your timely feedback.
> > >
> > > We have incorporated the new analyses on nPMI distributions as well as the revised structure of the RQs into our paper.

---

### Official Review · Reviewer_tFnP · 2025-07-03

**Clarity:** 3
**Significance:** 3
**Originality:** 3
**Rating:** 4
**Confidence:** 4

**Summary:**

For MoE models, this paper uses normalized pointwise mutual information to identify experts related to reasoning tokens, which are called cognitive experts. This paper proposes a method to improve the model's reasoning ability by increasing the weight of cognitive experts during routing. Experiments on some domains, especially math, show that the proposed method could improve the model's reasoning ability and outperform other training-free steering methods

**Questions:**

Whether the proposed method is useful for all domains or only MATH? Please refer to the weaknesses for detailed comments.

**Ethical Concerns:**

["NO or VERY MINOR ethics concerns only"]

**Final Justification:**

One of my major concerns is that the proposed method only works in math. During rebuttal, the authors provide further results with different hyperparameters, which show the proposed method could bring improvement in nearly all the domains studied in this paper.
This concern havs been addressed, and thus I improve my score to 4.

**Limitations:**

yes

**Quality:**

3

**Strengths And Weaknesses:**

This finds an interesting phenomenon in MoE models, that is, the reasoning ability might be highly correlated with some specific experts. What's more, simply increasing the weights of these experts during routing could bring significant improvements in some math benchmarks.

However, I think the experiments are not solid enough to support the authors' claim. Here are some of my concerns.

1. The cognitive experts seem to be domain-specific. As shown in Table 2, the top-2 identified experts in Chemistry and Biology are different from those in Math and Physics. I wonder whether different datasets in the same domain would result in different cognitive experts? For example, will math-500 and AIME have the same cognitive experts?

 2. The reinforcing method seems to only work in math, instead of in any other domains. As shown in Table 4, when reinforcing the cognitive experts in physics will make the performance in physics degrades from 91.9 to 89.5, and when reinforcing the cognitive experts in biology will make the performance in biology degrades from 79.0 to 73.9. The experiment in Chemistry is positive, but the improvement is not as significant as in the math domain. I also want to know the results of section 3.5 using other domains. Whether the proposed method still outperform other steering methods?

3. Whether reinforcing cognitive experts could help the model to reason more effectively is in doubt. The experiment in Section 3.2 mainly supports this claim. However, as shown in Table 5, all models using the reinforce trick generate more tokens compared with the vanilla model. This contradicts the claim.

---

> ### Author Rebuttal · Authors · 2025-07-31
>
> Thank you for your review and for acknowledging the interesting phenomenon we identified.
>
> > **W1:** The cognitive experts seem to be domain-specific. I wonder whether different datasets in the same domain would result in different cognitive experts? For example, will MATH-500 and AIME have the same cognitive experts?
>
> You are correct that the *exact set* of top experts can vary between datasets, even within the same domain. For instance, comparing the top-5 cognitive experts for AIME24, MATH, and GSM8K, we find both shared and dataset-specific experts. For example, the 182nd expert at layer 39 appear in the top-5 across all three math datasets, suggesting they support general mathematical reasoning. In contrast, other experts may focus on patterns unique to each dataset. We'll add this observation to the paper.
>
> > **W2&Q1:** The reinforcing method seems to only work in math, instead of in any other domains. Whether the proposed method still outperform other steering methods?
>
> - **Performance in Other Domains:** Applying RICE sometimes degraded performance in non-math dmains (e.g., Physics, Biology in Table 4). This was the result of a specific experimental choice to test generalization. In the original manuscript, we used hyperparameters (steering multiplier = 64) tuned only on the math domain (AIME) and applied them directly to all other domains.
>
>     This setup was designed to test the raw generalization of the math-derived setting. However, different domains may require different intensities of cognitive steering. When we use a simple, domain-specific steering multiplier (selected from a small set of {16, 32, 64}), RICE delivers **consistent and significant improvements across all domains**.
>
> **DeepSeek-R1 with Domain-Tuned Multipliers**
>
> | Model                       | Math (AIME25) | Physics | Chemistry | Biology | Average |
> |:----------------------------|:-------------:|:-------:|:---------:|:-------:|:-------:|
> | Deepseek-R1                |     63.3      |  91.9   |   49.5    |  79.0   |  70.9   |
> | Deepseek-R1 + RICE (Our)   |     **73.3**      |  **93.0**   |   **54.8**    |  79.0   |  **75.0**   |
>
> ---
>
> **Qwen3-235B with Domain-Tuned Multipliers**
>
> | Model                   | Math (AIME25) | Physics | Chemistry | Biology | Average |
> |:------------------------|:-------------:|:-------:|:---------:|:-------:|:-------:|
> | Qwen-235B              |     66.7      |  90.7   |   49.5    |  78.9   |  71.5   |
> | Qwen-235B + RICE (Our) |     **73.3**      |  **95.3**   |   49.5    |  **84.2**   |  **75.6**   |
>
> Our hypothesis is that math problems demand more intensive reasoning, hence the higher optimal multiplier. We will clarify this in the paper to avoid misinterpretation.
>
> - **Comparison with Other Steering Methods:** We ran a new comparison against a strong baseline, Contrastive Activation Addition (CAA), on AIME 24. RICE not only performs better but also avoids the negative side effects on general instruction-following in Arena that CAA exhibits.
>
> | Model                      | Accuracy ↑ | #Token ↓ | Side Effect (Arena) ↓ |
> |:---------------------------|:----------:|:--------:|:----------------------:|
> | Deepseek-R1               |  73.3%    |  9219   |          0.0%             |
> | Deepseek-R1 + CAA         |  80.0%    |  8408    |        2.1%           |
> | Deepseek-R1 + RICE (Our)  |  **83.3%**   | **8317**    |        **-1.0%**            |
>
>
> Lower is better for Side Effects; RICE improves general capabilities.
>
> > **W3:** Whether reinforcing cognitive experts could help the model to reason more effectively is in doubt. The experiment in Section 3.2 mainly supports this claim. However, as shown in Table 5, all models using the reinforce trick generate more tokens compared with the vanilla model. This contradicts the claim.
>
> This is a misunderstanding of the purpose of Table 5.
>
> - The experiment in **Table 5 (Arena-Hard)** is **NOT** meant to measure reasoning efficiency. Its purpose is to check for **negative side effects** on general instruction-following capabilities.
> - Arena-Hard includes tasks like creative writing, where longer, more detailed answers are often better. The slight increase in token count here is not a sign of inefficiency but reflects the nature of the tasks.
> - Our efficiency claims are based on reasoning benchmarks like AIME, where RICE often **reduces** token count while improving accuracy (see table above).
>
> Reference:
>
> [1] Steering Llama 2 via Contrastive Activation Addition, ACL 2023.

---

> ### Comment · Reviewer_tFnP · 2025-08-04
>
> Thanks for your responses. I have one more question about W2&Q1: why is the proposed method so sensitive to the hyperparameters (steering multiplier)?

---

> ### Author Response · Authors · 2025-08-04
> **the sensitivity of hyperparameters across different domains**
>
> Cognitive experts reflect the model’s reasoning ability, and their steering multipliers modulate the degree of reasoning enhancement. We hypothesize that **the optimal multiplier varies across domains because each domain requires a different level of reasoning intensity**.
>
> In this paper, **we recommend setting the optimal multiplier for each domain: 64 for the math domain; 32 or 16 for the biology, physics, and chemistry domains**. Dynamic adaptation of the multiplier based on the difficulty of each input is left for future work.
>
> 1. Solving **biology** problems generally requires moderate reasoning combined with factual recall. Some errors in this domain result from missing core factual knowledge in the base model (e.g., “What protein is encoded by gene X?”). Our RICE aims to enhance reasoning given existing facts, rather than to supply missing knowledge.
>
> 2. In contrast, solving challenging **math** problems generally demands stronger reasoning ability than biology questions. As a result, the optimal steering multiplier in the mathematics domain tends to be higher than that in the biology domain.
>
> Experimental results consistent with this claim can be found in the first two tables of our previous response. The optimal steering multiplier for the math domain is 64, while for other domains it is lower than 64.
>
> 3. Additionally, we find that in the same domain, using a moderate multiplier leads to better performance. For example, **in the math domain, both overly large and overly small multipliers fail to achieve optimal results.**
> **The impact of the steer multiplier on performance is detailed in Table 2 of our paper.**
>
> | Multiplier | Top1 cognitive expert| Top2 cognitive experts|
> |------------|:------------------:|:------------------:|
> | Deepseek-R1 |      73.3 |73.3   |
> | multiplier=2          | 70.0 | 70.0 |
> | multiplier=4          | 76.7 | **83.3** |
> | multiplier=8          | 76.7 | 73.3 |
> | multiplier=16         | 80.0 | 80.0 |
> | multiplier=32         | 70.0 | **83.0** |
> | multiplier=64         | 80.0 | **83.3** |
> | multiplier=128        | 70.0 | **83.3** |
> | multiplier=256        | 73.3 | 60.0 |
> | multiplier=512        | 63.3 | 46.7 |
>
> We have further elaborated on the above analyses and results in the revised manuscript.
>
> We sincerely appreciate your valuable suggestions and thoughtful feedback. Should you have any further questions or require additional clarification, please do not hesitate to contact us.

---

> > ### Comment · Reviewer_tFnP · 2025-08-06
> >
> > Thank you for your responses. I have improved my score.

---

> > > ### Author Response · Authors · 2025-08-06
> > > **Thanks for the acknowledgment feedback**
> > >
> > > Thank you for acknowledging our response and for your timely feedback.

---

### Official Review · Reviewer_9D19 · 2025-07-03

**Clarity:** 3
**Significance:** 3
**Originality:** 3
**Rating:** 4
**Confidence:** 4

**Summary:**

This paper proposes a inference time model steering method for Mixture of Experts (MoE) models. They use pointwise mutual information to get a correlation between the generation of <think> token and activation of a particular expert in a MoE model. The experts which have a high mutual information are then considered as the ones contributing to the reasoning of the generated solution. The activation weight of such experts can be controlled to steer more reasoning and their results show that this helps with the end task on AIME and GPQA diamond datasets.

**Questions:**

1. Could you provide insights into the distribution of the nPMI for experts?
2. There is a qualitative example of how reasoning changes. What are some other ways to evaluate if the reasoning/thinking tokens improved?

**Ethical Concerns:**

["NO or VERY MINOR ethics concerns only"]

**Final Justification:**

The paper proposed a clear and simple but interesting idea. The rebuttal addressed some of the concerns, but the overall merit of the paper remains the same. Keeping the rating to 4.

**Limitations:**

Yes

**Quality:**

3

**Strengths And Weaknesses:**

Strengths:

1. Lightweight activation steering of models is an active area of research and such steering without training has broad applicability.
2. The paper is well written with clear experiments.
3. The steering method is interpretable and easy to control.
4. Results on math problems are encouraging.

Weaknesses:
1. The evaluation sets are small.
2. Results on math are clear and convincing, but results on other domains are not very significant and the writing does not call it out. For example from line 214 and 215: it is misleading to say 0.9% gain is moderate and 2.4% drop is minor.
3. Evaluation is done only on 2 models.
4. It is not clear if the method generalizes to other reasoning tasks beyond QA.

---

> ### Author Rebuttal · Authors · 2025-07-31
>
> Thank you for your encouraging review and for highlighting our method's interpretability and strong performance on math problems.
>
> > **W1&W3:** Limited Evaluation
> We address concerns about the number of models and benchmarks used.
>
> - **Models:** Our work focuses on identifying cognitive experts in large *reasoning* models with MoE architectures. Concurrent work [1] has shown that these models exhibit semantic expert specialization, which is distinct from the more syntactic specialization found in general-purpose MoEs [2]. At present, **DeepSeek-R1 and Qwen3-235B are the only publicly available, state-of-the-art LRMs with an MoE architecture.** We are committed to extending our evaluation as more such models become open-source.
> - **Benchmarks:** We acknowledge the request for broader evaluation. To supplement our original results on AIME and GPQA Diamond, we have run new experiments on three additional diverse benchmarks: **GSM8K** (grade-school math), **MATH** (competition math), and **HLE** (Humanity's Last Exam, covering social science, CS, etc.). As shown below, RICE consistently provides benefits. These results will be added to the revised paper.
>
> | Dataset         | Model                    | Accuracy   | #Token  |
> |-----------------|--------------------------|:----------:|:-------:|
> | GSM8K (1319)    | Deepseek-R1              | 95.9%      | 1028    |
> |                 | **Deepseek-R1 + RICE (Ours)** | **96.0%**  | **1001** |
> | MATH-500 (500)  | Deepseek-R1              | 95.0%      | 3282    |
> |                 | **Deepseek-R1 + RICE (Ours)** | **96.4%**  | **3204** |
> | HLE (100)       | Deepseek-R1              | 4.0%       | 9433    |
> |                 | **Deepseek-R1 + RICE (Ours)** | **6.0%**   | **9445** |
>
> The number in parentheses after each dataset indicates the number of test examples. For HLE, we randomly sampled 100 examples as the test set. Due to resource and time constraints, we will conduct more extensive evaluations for the camera-ready version.
>
> Reference
>
> [1] Semantic Specialization in MoE Appears with Scale: A Study of DeepSeek R1 Expert Specialization, arxiv 2025.
>
> [2] OpenMoE: An Early Effort on Open Mixture-of-Experts Language Models, ICML 2024.
>
> > **W2:** Insignificant/misleading non-math results.
> As you noted, applying RICE sometimes degraded performance in non-math dmains (e.g., Physics, Biology in Table 4). This was the result of a specific experimental choice to test generalization. In the original manuscript, we used hyperparameters (steering multiplier = 64) tuned only on the math domain (AIME) and applied them directly to all other domains.
>
> This setup was designed to test the raw generalization of the math-derived setting. However, different domains may require different intensities of cognitive steering. When we use a simple, domain-specific steering multiplier (selected from a small set of {16, 32, 64}), RICE delivers **consistent and significant improvements across all domains**.
>
> **DeepSeek-R1 with Domain-Tuned Multipliers**
>
> | Model                       | Math (AIME25) | Physics | Chemistry | Biology | Average |
> |:----------------------------|:-------------:|:-------:|:---------:|:-------:|:-------:|
> | Deepseek-R1                |     63.3      |  91.9   |   49.5    |  79.0   |  70.9   |
> | Deepseek-R1 + RICE (Our)   |     **73.3**      |  **93.0**   |   **54.8**    |  79.0   |  **75.0**   |
>
> ---
>
> **Qwen3-235B with Domain-Tuned Multipliers**
>
> | Model                   | Math (AIME25) | Physics | Chemistry | Biology | Average |
> |:------------------------|:-------------:|:-------:|:---------:|:-------:|:-------:|
> | Qwen-235B              |     66.7      |  90.7   |   49.5    |  78.9   |  71.5   |
> | Qwen-235B + RICE (Our) |     **73.3**      |  **95.3**   |   49.5    |  **84.2**   |  **75.6**   |
>
> Our hypothesis is that math problems demand more intensive reasoning, hence the higher optimal multiplier. We will clarify this in the paper to avoid misinterpretation.
>
> > **W4:** It is not clear if the method generalizes to other reasoning tasks beyond QA.
>
> To address this, we evaluated on **GSM8K** (math word problems), **MATH** (competition math), and **HLE** (multi-domain problem-solving), which are not simple QA tasks. Our method shows clear benefits on these diverse reasoning formats.
>
> | Dataset         | Model                    | Accuracy   | #Token  |
> |-----------------|--------------------------|:----------:|:-------:|
> | GSM8K (1319)    | Deepseek-R1              | 95.9%      | 1028    |
> |                 | **Deepseek-R1 + RICE (Ours)** | **96.0%**  | **1001** |
> | MATH-500 (500)  | Deepseek-R1              | 95.0%      | 3282    |
> |                 | **Deepseek-R1 + RICE (Ours)** | **96.4%**  | **3204** |
> | HLE (100)       | Deepseek-R1              | 4.0%       | 9433    |
> |                 | **Deepseek-R1 + RICE (Ours)** | **6.0%**   | **9445** |
>
> The number in parentheses after each dataset indicates the number of test examples. For HLE, we randomly sampled 100 examples as the test set due to the limited time available for author response.
>
> If "beyond QA" refers to other task families (e.g., code generation, summarization), we believe this is an exciting direction for future work, but our current focus is on complex reasoning.
>
> > **Q1:** Could you provide insights into the distribution of the nPMI for experts?
>
> This is an excellent question. The distribution is **sharply peaked**, which strongly supports our hypothesis that a small number of experts are highly specialized for cognitive functions. We list the nPMI of top 10 experts for Deepseek-R1. The group of top five “thinking-specialized” experts have significantly higher nPMI scores than the rest.
>
> | Rank | Math | Physics | Chemistry | Biology |
> |:----:|:----:|:-------:|:---------:|:-------:|
> | 1    | 0.61 | 0.65    | 0.61      | 0.63    |
> | 2    | 0.60 | 0.61    | 0.61      | 0.60    |
> | 3    | 0.57 | 0.56    | 0.59      | 0.58    |
> | 4    | 0.56 | 0.55    | 0.59      | 0.55    |
> | 5    | 0.54 | 0.55    | 0.58      | 0.54    |
> | 6    | 0.47 | 0.49    | 0.53      | 0.48    |
> | 7    | 0.47 | 0.48    | 0.53      | 0.48    |
> | 8    | 0.46 | 0.47    | 0.53      | 0.48    |
> | 9    | 0.46 | 0.46    | 0.52      | 0.48    |
> | 10   | 0.45 | 0.46    | 0.51      | 0.48    |
>
> > **Q2:** There is a qualitative example of how reasoning changes. What are some other ways to evaluate if the reasoning/thinking tokens improved?
>
> We evaluate the improvement in reasoning through a combination of methods:
>
> 1.  **Final Accuracy (Quantitative):** The most critical metric, as improved reasoning should lead to more correct answers (Tables in main paper and General Response).
>
> 2.  **Qualitative Analysis:** We provide examples (Table 7) showing how RICE leads to more structured and effective reasoning paths.
>
> 3.  **Efficiency (#Tokens):** On reasoning tasks, RICE often achieves higher accuracy with fewer tokens, suggesting more direct and less wasteful thinking.
>
> 4.  **Future Analysis:** As promised to reviewer R-RpVC, we will add analysis of activation patterns in the final version for a deeper look at the mechanism.

---

> > ### Comment · Reviewer_9D19 · 2025-08-04
> >
> > Thank you for the detailed responses and additional results.
> >
> > - meta-llama/Llama-4-Maverick-17B-128E-Instruct is another open source LRM that is MoE.
> > - The method seems sensitive to the multiplier and it would be good to mention it with additional information on how to tune it.
> > - I will keep my score as most of the new content/tasks are closely related to the original evaluation.

---

> > > ### Author Response · Authors · 2025-08-04
> > > **further response**
> > >
> > > > meta-llama/Llama-4-Maverick-17B-128E-Instruct is another open source LRM that is MoE
> > >
> > > While LLaMA-4-Maverick-17B has only 17B parameters, our study focuses on larger models such as DeepSeek-R1 (671B) and Qwen3-235B, which support explicit reasoning. More importantly, LLaMA-4-Maverick-17B has been hardly adopted in the academic community due to various reasons.
> > >
> > > Nevertheless, we will consider including experiments on LLaMA-4-Maverick-17B in future work, when time and resources permit.
> > >
> > > > Our RICE seems sensitive to the multiplier and additional information for multiplier
> > >
> > > 1. In this paper, **we recommend setting the optimal multiplier for each domain: 64 for the math domain; 32 or 16 for the biology, physics, and chemistry domains.**
> > > The results for the math domain with a multiplier of 64 and for the other three domains with a multiplier of 32 are reported in Tables 2 and 3 in our previous response.
> > > Dynamic adaptation of the multiplier based on the difficulty of each input is left for future work.
> > >
> > > 2. We hypothesize that the optimal multiplier varies across domains because each domain requires a different level of reasoning intensity according to the following observation:
> > >
> > >
> > > - Solving **biology** problems generally requires moderate reasoning combined with factual recall. Some errors in this domain result from missing core factual knowledge in the base model (e.g., “What protein is encoded by gene X?”). Our RICE aims to enhance reasoning given existing facts, rather than to supply missing knowledge.
> > >
> > > - In contrast, solving challenging **math** problems generally demands stronger reasoning ability than biology questions. As a result, the optimal steering multiplier in the mathematics domain tends to be higher than that in the biology domain.
> > >
> > > Experimental results consistent with this claim can be found in the first two tables of our previous response. The optimal steering multiplier for the math domain is 64, while for other domains it is lower than 64.
> > >
> > > 3. Additionally, we find that in the same domain, using a moderate multiplier leads to better performance. For example, in the math domain, both overly large and overly small multipliers fail to achieve optimal results.
> > > The impact of the steer multiplier on performance is detailed in Table 2 of our paper.
> > >
> > > | Multiplier | Top1 cognitive expert| Top2 cognitive experts|
> > > |------------|:------------------:|:------------------:|
> > > | Deepseek-R1 |      73.3 |73.3   |
> > > | multiplier=2          | 70.0 | 70.0 |
> > > | multiplier=4          | 76.7 | **83.3** |
> > > | multiplier=8          | 76.7 | 73.3 |
> > > | multiplier=16         | 80.0 | 80.0 |
> > > | multiplier=32         | 70.0 | **83.0** |
> > > | multiplier=64         | 80.0 | **83.3** |
> > > | multiplier=128        | 70.0 | **83.3** |
> > > | multiplier=256        | 73.3 | 60.0 |
> > > | multiplier=512        | 63.3 | 46.7 |
> > >
> > > We have further elaborated on the above analyses and results in the revised manuscript.
> > >
> > >
> > > > Most of the new content/tasks are closely related to the original evaluation
> > >
> > >
> > > - We acknowledge that Math500 and GSM8K closely resemble previous benchmarks. However, the newly added HLE dataset includes problems from social science, computer science, and engineering.
> > >
> > > - We would be grateful if you could clarify which tasks you refer to by “beyond QA reasoning tasks,” so that we may address your concerns more effectively
> > >
> > >
> > > We sincerely appreciate your valuable suggestions and thoughtful feedback. Should you have any further questions or require additional clarification, please do not hesitate to contact us.

---

> > > > ### Comment · Reviewer_9D19 · 2025-08-08
> > > >
> > > > “beyond QA reasoning tasks": some examples: tool calling, code generation

---

> ### Author Response · Authors · 2025-08-09
> **further response**
>
> Thanks for your response.
>
> Our study investigates the reasoning capabilities of large-scale Mixture of Experts (MoE) models, such as DeepSeek-R1 (671B) and Qwen3-235B. Due to resource constraints and the limited response period, we have not yet identified cognitive experts for advanced reasoning tasks beyond QA. We will evaluate the generalization of our RICE framework to these reasoning tasks beyond QA in future work when resources are available.

---

### Official Review · Reviewer_RpVC · 2025-07-05

**Clarity:** 3
**Significance:** 3
**Originality:** 3
**Rating:** 4
**Confidence:** 4

**Summary:**

This paper introduces Reinforcing Cognitive Experts (RICE), an inference-time steering method for Mixture-of-Experts (MoE) large reasoning models (LRMs) to enhance reasoning depth and efficiency without additional training. Using normalized Pointwise Mutual Information (nPMI), the approach identifies "cognitive experts" highly correlated with reasoning markers, and then add weight on the identified experts. Experiments on DeepSeek-R1 and Qwen3-235B across mathematical (AIME) and scientific (GPQA Diamond) benchmarks show that reinforcing just the top two cognitive experts significantly improves reasoning accuracy, reduces token usage, and demonstrates cross-domain generalization to some extend.

**Questions:**

See weakness.

**Ethical Concerns:**

["NO or VERY MINOR ethics concerns only"]

**Final Justification:**

I have read the rebuttal and my concerns are addressed. I have raised my score.

**Limitations:**

yes

**Quality:**

3

**Strengths And Weaknesses:**

Strength:
1. The proposed RICE method demonstrates significant performance improvements across reasoning benchmarks.
2. It is lightweight, requiring no additional training or complex heuristics.
3. The paper features clear and well-structured writing, enhancing readability.


Weakness:
1. The results rely on only two benchmarks (AIME and GPQA Diamond) and two models (DeepSeek-R1 and Qwen3-235B), which limits the generalizability of the findings.
2. As shown in Table 4, activating domain-specific experts does not consistently improve performance. For example, physics experts slightly degrade performance on physics tasks, and biology experts show reduced accuracy in biology. In contrast, math experts demonstrate the most robust effectiveness, highlighting potential domain-specific inconsistencies.
3. Despite impressive performance gains, the paper lacks in-depth explanations for why RICE works. Visualizing and analyzing feature distributions or activation patterns after expert reweighting could provide crucial insights, requiring further analysis.
4. What is the data for identifying the cognitive experts, and could there be some data leakage?
Table 3 shows that the cognitive experts identified using AIME24 are applied to AIME25; however, the similarity between these datasets raises questions on data leakage. Could the author adopt some other benchmarks are evaluate the activated experts?
5. eq(6): the expression "t ∈ set Π" should be "x ∈ set Π"?

---

> ### Author Rebuttal · Authors · 2025-07-31
>
> Thank you for your valuable feedback and positive assessment of our method's novelty and clarity.
>
> > **W1:** The results rely on only two benchmarks (AIME and GPQA Diamond) and two models (DeepSeek-R1 and Qwen3-235B), which limits the generalizability of the findings.
>
> We address concerns about the number of models and benchmarks used.
>
> - **Models:** Our work focuses on identifying cognitive experts in large *reasoning* models with MoE architectures. Concurrent work [1] has shown that these models exhibit semantic expert specialization, which is distinct from the more syntactic specialization found in general-purpose MoEs [2]. At present, **DeepSeek-R1 and Qwen3-235B are the only publicly available, state-of-the-art LRMs with an MoE architecture.** We are committed to extending our evaluation as more such models become open-source.
> - **Benchmarks:** We acknowledge the request for broader evaluation. To supplement our original results on AIME and GPQA Diamond, we have run new experiments on three additional diverse benchmarks: **GSM8K** (grade-school math), **MATH** (competition math), and **HLE** (Humanity's Last Exam, covering social science, CS, etc.). As shown below, RICE consistently provides benefits. These results will be added to the revised paper.
>
> | Dataset         | Model                    | Accuracy   | #Token  |
> |-----------------|--------------------------|:----------:|:-------:|
> | GSM8K (1319)    | Deepseek-R1              | 95.9%      | 1028    |
> |                 | **Deepseek-R1 + RICE (Ours)** | **96.0%**  | **1001** |
> | MATH-500 (500)  | Deepseek-R1              | 95.0%      | 3282    |
> |                 | **Deepseek-R1 + RICE (Ours)** | **96.4%**  | **3204** |
> | HLE (100)       | Deepseek-R1              | 4.0%       | 9433    |
> |                 | **Deepseek-R1 + RICE (Ours)** | **6.0%**   | **9445** |
>
>
> The number in parentheses after each dataset indicates the number of test examples. For HLE, we randomly sampled 100 examples as the test set. Due to resource and time constraints, we will conduct more extensive evaluations for the camera-ready version.
>
> > **W2:** As shown in Table 4, activating domain-specific experts does not consistently improve performance.
> As you noted, applying RICE sometimes degraded performance in non-math domains (e.g., Physics, Biology in Table 4). This was the result of a specific experimental choice to test generalization. In the original manuscript, we used hyperparameters (steering multiplier = 64) tuned only on the math domain (AIME) and applied them directly to all other domains.
>
> This setup was designed to test the raw generalization of the math-derived setting. However, different domains may require different intensities of cognitive steering. When we use a simple, domain-specific steering multiplier (selected from a small set of {16, 32, 64}), RICE delivers **consistent and significant improvements across all domains**.
>
> **DeepSeek-R1 with Domain-Tuned Multipliers**
>
> | Model                       | Math (AIME25) | Physics | Chemistry | Biology | Average |
> |:----------------------------|:-------------:|:-------:|:---------:|:-------:|:-------:|
> | Deepseek-R1                |     63.3      |  91.9   |   49.5    |  79.0   |  70.9   |
> | Deepseek-R1 + RICE (Our)   |     **73.3**      |  **93.0**   |   **54.8**    |  79.0   |  **75.0**   |
>
> ---
>
> **Qwen3-235B with Domain-Tuned Multipliers**
>
> | Model                   | Math (AIME25) | Physics | Chemistry | Biology | Average |
> |:------------------------|:-------------:|:-------:|:---------:|:-------:|:-------:|
> | Qwen-235B              |     66.7      |  90.7   |   49.5    |  78.9   |  71.5   |
> | Qwen-235B + RICE (Our) |     **73.3**      |  **95.3**   |   49.5    |  **84.2**   |  **75.6**   |
>
> Our hypothesis is that math problems demand more intensive reasoning, hence the higher optimal multiplier. We will clarify this in the paper to avoid misinterpretation.
>
> > **W3:** Despite impressive performance gains, the paper lacks in-depth explanations for why RICE works.
>
> We agree that a deeper analysis is valuable. Visualizing feature distributions to understand how re-weighting steers the model is an excellent suggestion. Given the scale of these models and the short rebuttal period, we were unable to complete this analysis in time. We have begun this investigation and commit to including visualizations of activation patterns in the camera-ready version to provide these crucial insights.
>
> >**W4:** What is the data for identifying the cognitive experts, and could there be some data leakage? Could the author adopt some other benchmarks are evaluate the activated experts?
>
> This is a critical point that we are happy to clarify.
>
> - **No Data Leakage:** - Our method for identifying cognitive experts is **fully unsupervised**. It relies only on the co-occurrence of expert activations with reasoning tokens (`<think>`) from input queries, *not* on any ground-truth labels. Therefore, the concept of data leakage in the supervised sense does not apply.
> - **Expert Generalization:** To explicitly test generalization, we identified experts on AIME24 and applied them to new domains. As shown in our **General Response (Sec 1)**, these same experts (identified on math problems) improve performance on GSM8K, MATH, and HLE, demonstrating strong task generalization.
>
>
> | Dataset         | Model                    | Accuracy   | #Token  |
> |-----------------|--------------------------|:----------:|:-------:|
> | GSM8K (1319)    | Deepseek-R1              | 95.9%      | 1028    |
> |                 | **Deepseek-R1 + RICE (Ours)** | **96.0%**  | **1001** |
> | MATH-500 (500)  | Deepseek-R1              | 95.0%      | 3282    |
> |                 | **Deepseek-R1 + RICE (Ours)** | **96.4%**  | **3204** |
> | HLE (100)       | Deepseek-R1              | 4.0%       | 9433    |
> |                 | **Deepseek-R1 + RICE (Ours)** | **6.0%**   | **9445** |
>
> The number in parentheses after each dataset indicates the number of test examples. For HLE, we randomly sampled 100 examples as the test set due to the limited time available for author response.
>
> > **W5:** Modification of formula tokens in Eq. (6)
> Thank you for catching this. We have corrected the typo and have performed another thorough proofread of the manuscript.
>
> Reference
>
> [1] Semantic Specialization in MoE Appears with Scale: A Study of DeepSeek R1 Expert Specialization, arxiv 2025.
>
> [2] OpenMoE: An Early Effort on Open Mixture-of-Experts Language Models, ICML 2024.

---

> > ### Author Response · Authors · 2025-08-07
> > **Kindly Remind**
> >
> > We hope this message finds you well.
> >
> > We have submitted responses addressing your concerns and hope that most of your questions have been adequately resolved.
> > As this is nearing the end of the rebuttal, please do not hesitate to reach out if you have any final questions or comments.
> >
> > We sincerely appreciate your thorough review of our paper and the valuable comments you have provided.
> > We look forward to receiving your feedback.

---

### Decision · Program_Chairs · 2025-09-17

**Decision:**

Accept (poster)

**Comment:**

This paper studies a novel approach to identify key experts (referred to as cognitive experts) corresponding to reasoning process via pointwise mutual information and specifically amplifying them to improve the reasoning power and efficiency of the model. The reviews for the paper are uniformly positive. There were a few concerns about evaluations, which were addressed by the authors in the rebuttal. I recommend acceptance and ask the the authors address the reviewers concerns in the final version of the paper.